# Vitamin B_12_ in Foods, Food Supplements, and Medicines—A Review of Its Role and Properties with a Focus on Its Stability

**DOI:** 10.3390/molecules28010240

**Published:** 2022-12-28

**Authors:** Žane Temova Rakuša, Robert Roškar, Neal Hickey, Silvano Geremia

**Affiliations:** 1Faculty of Pharmacy, University of Ljubljana, 1000 Ljubljana, Slovenia; 2Department of Chemical and Pharmaceutical Sciences, Centre of Excellence in Biocrystallography, University of Trieste, Via L. Giorgieri 1, 34127 Trieste, Italy

**Keywords:** 5′-deoxyadenosylcobalamin, cyanocobalamin, food supplements, fortified foods, hydroxocobalamin, medicines, methylcobalamin

## Abstract

Vitamin B_12_, also known as the anti-pernicious anemia factor, is an essential micronutrient totally dependent on dietary sources that is commonly integrated with food supplements. Four vitamin B_12_ forms—cyanocobalamin, hydroxocobalamin, 5′-deoxyadenosylcobalamin, and methylcobalamin—are currently used for supplementation and, here, we provide an overview of their biochemical role, bioavailability, and efficacy in different dosage forms. Since the effective quantity of vitamin B_12_ depends on the stability of the different forms, we further provide a review of their main reactivity and stability under exposure to various environmental factors (e.g., temperature, pH, light) and the presence of some typical interacting compounds (oxidants, reductants, and other water-soluble vitamins). Further, we explore how the manufacturing process and storage affect B_12_ stability in foods, food supplements, and medicines and provide a summary of the data published to date on the content-related quality of vitamin B_12_ products on the market. We also provide an overview of the approaches toward their stabilization, including minimization of the destabilizing factors, addition of proper stabilizers, or application of some (innovative) technological processes that could be implemented and contribute to the production of high-quality vitamin B_12_ products.

## 1. Introduction

Vitamin B_12_ is an essential micronutrient that belongs to the group of water-soluble B-complex vitamins. In a narrow sense, the term vitamin B_12_ implies cyanocobalamin, the anti-pernicious anemia factor isolated and crystallized in Glaxo laboratories and structurally characterized by the Nobel Prize winner Dorothy Hodgkin in the early 1950s. This is the primary vitamin B_12_ form used for food supplementation purposes, however, more broadly, the term includes a group of corrinoid compounds known as cobalamins (Cbls) [1,2,3]. These include the two Cbl forms with biological activity: 5′-deoxyadenosylcobalamin (AdoCbl) and methylcobalamin (MeCbl), as well as other forms, such as cyanocobalamin (CNCbl), hydroxocobalamin/aquocobalamin (OHCbl/H_2_OCbl^+^), and analogous Cbl derivates (glutathionyl-, nitrite-, nitroso-, sulfito-, etc.) [1,3,4,5].

The major dietary source of vitamin B_12_ is animal-derived food, i.e., meat, milk, eggs, and fish [6,7]. Plant foods typically do not contain substantial vitamin B_12_ amounts, with some exceptions (dried green and purple lavers) [6,8]. Therefore, vegetarians and vegans are especially prone to vitamin B_12_ deficiency; vitamin B_12_ supplementation through food supplements and fortified food is recommended to prevent deficiency in these populations [9,10]. On the other hand, vitamin B_12_ in the form of prescribed medicines is mainly used for the prevention and treatment of vitamin B_12_ deficiency [9]. Vitamin B_12_ medicines, food supplements, and fortified foods most commonly contain CNCbl, due to its better stability compared to other vitamin B_12_ forms [1,3,11]. Alternatively, OHCbl may also be found in medicines and foods. In accordance with the European Commission Regulation No 1170/2009 of 30 November 2009, CNCbl, OHCbl, AdoCbl, and MeCbl may be used in the manufacture of food supplements, whereas CNCbl and OHCbl may also be added to foods [12]. The stability of each form is thus an important factor to consider during the formulation and manufacture of quality medicines, food supplements, and fortified foods and is also associated with their efficient and safe use. Therefore, the aim of this manuscript is to provide a review of vitamin B_12_’s activity and use, with a special focus on its stability and approaches for the stabilization of its frequently used forms in foods, food supplements, and medicines. Aspects related to its absorption, distribution, metabolism, excretion (ADME), activation, and stability in complexes with its transport proteins in the human body, which all affect the clinical outcomes and efficiency of different cobalamines supplements, will also be reported.

## 2. Vitamin B_12_ ADME Properties, Activation, and Stability in the Human Body

Dietary vitamin B_12_ is usually bound to food proteins [13,14]. Its active absorption begins with its release from food, which is facilitated by the low pH in the gastric lumen (Figure 1) [13,15]. The liberated vitamin B_12_ is immediately bound to haptocorrin (transcobalamin-I) (HC), a vitamin B_12_-binding glycoprotein produced by salivary and oesophageal glands, which protects it from acid hydrolysis in the stomach [15,16,17]. The vitamin B_12_-HC complex is degraded by pancreatic proteases in the duodenum, thus releasing vitamin B_12_, which further binds to intrinsic factor (IF) produced by the parietal cells of the stomach [14,17,18]. The vitamin B_12_-IF complex then passes to the distal ileum, which is its primary absorption site [15,17]. This complex binds to its enterocyte membrane receptor and enters the enterocytes by endocytosis. A small portion of the ingested vitamin B_12_ (1–2%) is absorbed by the less efficient passive diffusion, which becomes more important in patients with a deficiency or lack of IF (e.g., gastrectomized patients), as well as in the event of exceeded capacity of the IF system [1,15]. The capacity of the IF system to absorb vitamin B_12_ is limited and it decreases with the application of larger vitamin B_12_ doses, ranging from 80% for 0.1 μg oral dose to 50% for 1 μg oral dose and <10% in doses near or above 20 μg. High vitamin B_12_ oral doses (≥500 μg) result in about 1% absorption [15,16,19]. Passive vitamin B_12_ absorption also occurs in the nasal mucous membranes after its nasal application [15]. The available literature indicates different vitamin B_12_ bioavailability after nasal application, depending on its applied form, and is estimated to be 2–5% for OHCbl [20], 2–6% for CNCbl [21], and ~20% for MeCbl [22].

Within the enterocytes, lysosomes degrade IF and the free vitamin B_12_ binds to transcobalamin-II (TC-II), a nonglycosylated protein, which transports vitamin B_12_ in the blood stream [14,17,18]. The vitamin B_12_-TC-II complex has a half-life of 60–90 min and is quickly taken up by the target cells [23]. The cellular uptake of the vitamin B_12_-TC-II complex proceeds as receptor-mediated endocytosis, followed by its intracellular degradation in the lysosomes and the release of vitamin B_12_. Within the target cell, vitamin B_12_, regardless of its ingested form, is converted to its two active forms, AdoCbl and MeCbl, by a complex intracellular pathway in which different chaperones and transporters are involved [24,25,26,27]. The methylmalonic aciduria and homocystinuria type C protein (MMACHC) is the main chaperone, which binds the released vitamin B_12_ from the lysosomes in a base-off conformation (the ligand 5’,6´-dimethylbenzimidazole in vitamin B_12_ is replaced with a histidine residue of the protein). The MMACHC also catalyzes the conversion of any vitamin B_12_ form into the cob(II)alamin intermediate by the decyanation of CNCbl and dealkylation of alkylCbls, which also requires the activity of glutathione S-transferase. The following step of cob(II)alamin conversion into AdoCbl and MeCbl involves the activity of various enzymes, including MMACHC, methylmalonic aciduria and homocystinuria type D protein (MMADHC), and methionine synthase reductase [25,26,27,28]. AdoCbl and MeCbl exhibit their physiological effects upon binding to their target enzymes methionine synthase (MetH), and methylmalonyl coenzyme A mutase (MMCA) (Figure 1) [14,15,18]. Thus, MeCbl is the predominant vitamin B_12_ form in the plasma and AdoCbl in all tissues [1].

Excess vitamin B_12_ is accumulated in the liver and kidney. The average vitamin B_12_ content in heathy adults is estimated to be 2–3 mg, about half of which is stored in the liver [1,29]. Part of the vitamin B_12_, stored in the liver (0.5–5.0 μg vitamin B_12_/day), is secreted in the bile (Figure 1). A majority (50–80%) of it is reabsorbed, bound to IF, in the ileal enterocytes [1,15,17]. Enterohepatic circulation with vitamin B_12_ recycling is thus another important pathway to maintain adequate vitamin B_12_ levels in the body. This reabsorption is the reason for relatively low losses of B_12_ and the slow manifestation of vitamin B_12_ deficiency even in the absence of its intake [1]. The unabsorbed vitamin B_12_ from food or bile is predominantly excreted in the feces, with an estimated daily loss of approximately 0.1% of its body stores [1,30]. In situations where the circulating vitamin B_12_ in the blood exceeds its binding capacity (e.g., after injection), the excess amounts are excreted in the urine [1,30].

### Vitamin B_12_ Stability in the Complex with Transport Proteins in the Human Body 

Vitamin B_12_ forms a series of complexes with its transport proteins (HC, IF, TC-II, Figure 1). The structures of the transport proteins are very similar [31]. Each transport protein almost completely envelops and protects vitamin B_12_ (Figure 2). Cbls bind tightly to their transport proteins with K_d_ values in the subpicomolar range (TC-II-Cbl complex has a Kd of 5 × 10^−15^ M) [32]. The solvent-accessible area of the complexed Cbl is very low (7–17%, depending on the vitamin B_12_ form) and only the ribosyl CH_2_OH group and in some cases the *β*-ligand are exposed to solvent (Figure 2) [18]. Although many interactions between cobalamin and its binding site are conserved among the transporters, a significant difference of TC-II is the presence of a His residue, absent in HC and IF, which coordinates to the Co ion of Cbl in humans (after displacement of H_2_O at the upper axial position of H_2_O-Cbl) [31]. A protective and stabilizing role of TC-II during B_12_ transport in plasma was hypothesized. TC-II binding might be advantageous to impede the coordination of exogenous molecules to the Co ion, which causes the intracellular reductive conversion of Cbls to be more difficult [33]. Therefore, the most significant functional group unprotected by protein complexation is the 5′-hydroxyl group on the ribose of the nucleotide moiety of Cbl (Figure 2). This is the ideal site for the synthesis of vitamin B_12_ bioconjugates, which maintain a high affinity toward the transport protein in plasma [34].

## 3. Mechanism of Action of Vitamin B_12_ in the Human Body

In the human body, vitamin B_12_ acts as a cofactor of several vitamin B_12_-dependent enzymes, including methyltransferases, such as MetH, and mutases, such as MMCA [18,36]. The latter is an AdoCbl-dependent enzyme, involved in the metabolism of branched-chain amino acids, odd-chain fatty acids, and cholesterol [36,37]. MMCA catalyzes the isomerization of L-methylmalonyl-coenzyme A (CoA) into succinyl-CoA for further metabolic reactions within the tricarboxylic acid cycle [36,38]. Therefore, vitamin B_12_ deficiency leads to the accumulation and increased serum and urine concentration of methylmalonic acid, the precursor of L-methylmalonyl-CoA, which is utilized in the early diagnosis of vitamin B_12_ deficiency [39,40]. Improper isomerization of L-methylmalonyl-CoA into succinyl-CoA and the accumulation of methylmalonic acid cause the formation of non-physiological fatty acids, which are incorporated into neuronal lipids in the myelin sheaths. This results in impaired signal transmission and the various neurological and neuropsychiatric consequences associated with vitamin B_12_ deficiency (nerve atrophy, neuropathy, myelopathy, depression, dementia, etc.) [36,37,39,41]. MetH is a MeCbl-dependent enzyme, involved in the methionine cycle, in which the active folate form 5-methyltetrahydrofolate transfers a methyl group to homocysteine, generating methionine and tetrahydrofolate [36,42,43]. This reaction is thus of particular importance not only for the formation of the essential amino acid methionine but also for its further conversion to S-adenosylmethionine (SAM). The latter is a universal methyl donor in a variety of vital physiological processes including the methylation of DNA, RNA, histones, and other proteins [36,43,44].

## 4. Vitamin B_12_ Deficiency

Vitamin B_12_ deficiency is an important public health issue, affecting an estimated 6% of the worldwide population and between 1.6% and 10% of the European populations [45]. The prevalence of vitamin B_12_ deficiency is generally higher among the elderly [46,47], vegetarians and vegans [48,49], pregnant women [50,51], infants breastfeeding from mothers with B_12_ deficiency [13,46], and, in general, in developing countries [45,51]. Due to its small losses, as a result of enterohepatic circulation, and its high amount of storage in the body (on average between 2 and 3 mg in healthy adults), vitamin B_12_ deficiency caused by its insufficient intake usually clinically manifests after a few years [1]. However, its deficiency becomes apparent more rapidly in the case of vitamin B_12_ malabsorption (e.g., due to pernicious anemia, food-bound vitamin B_12_ malabsorption, celiac disease, inflammatory bowel disease, Whipple disease), especially if combined with its low dietary intake [16]. Other common causes of vitamin B_12_ deficiency include inherited disorders (TC-II deficiency), the use of some medications (e.g., cholestyramine, metformin), and chronic alcoholism [9,13,39,46]. Considering the above-discussed actions of vitamin B_12_ in the human body, its deficiency affects important processes requiring intense cell replication, such as erythropoiesis. Therefore, vitamin B_12_ deficiency typically manifests clinically with hematological (megaloblastic anemia) and neurological symptoms (irritability, dementia, depression, visual disturbances, tingling or numbness of the extremities, paranoia, and psychosis) as well as some cellular and molecular consequences (cellular stress, apoptosis, and accumulation of homocysteine and methylmalonic acid) [9,39,46,52]. The diagnosis of severe vitamin B_12_ deficiency is typically based on hematological changes (elevated mean corpuscular volume of erythrocytes) and serum vitamin B_12_ concentration (<148 pmol/L (200 ng/L)). Other sensitive biomarkers of vitamin B_12_ deficiency include elevated plasma total homocysteine and methylmalonic acid levels and decreased serum vitamin B_12_-TC-II complex concentrations [1,46,53].

## 5. Treatment and Prevention of Vitamin B_12_ Deficiency

In healthy individuals, vitamin B_12_ daily intakes of 1–4 μg are considered sufficient to meet nutritional requirements [1]. Based on different biomarkers of vitamin B_12_ status and its mean intakes in the EU, the European Food Safety Authority (EFSA) Panel established adequate intakes for vitamin B_12_ at 4 μg/day for adults, 4.5 μg/day for pregnant women, and 5 μg/day for breastfeeding women. They also proposed estimated adequate intakes ranging between 1.5 μg/day for infants and 4 μg/day in children above 15 years [1]. The majority of the healthy population can achieve these values through the regular consumption of a mixed diet, including foods that are naturally rich in vitamin B_12_ (meat, milk and dairy products, fish, and eggs) [54]. However, dietary supplementation with vitamin B_12_ supplements is recommended in specific population groups, who are at higher risk for developing vitamin B_12_ deficiency (vegetarians, vegans, the elderly, and pregnant and lactating women) as the most efficient way of preventing its deficiency, even when vitamin B_12_ fortified foods are consumed [54,55,56,57]. A wide variety of vitamin B_12_ supplements may be found on the market in terms of the contained vitamin B_12_ form (mostly CNCbl due to its stability, but also MeCbl and OHCbl), its content (from a few μg up to a few mg), the presence of other micronutrients (supplements containing only vitamin B_12_ or in combinations with other B-complex vitamins and multivitamin/mineral supplements), and their dosage forms (tablets, effervescent tablets, drops, syrups, oral sprays, capsules, etc.) [54,58]. Contrary to dietary supplements, which are generally used for the prevention of vitamin B_12_ deficiency, high dose vitamin B_12_ treatments are usually indicated and used in conditions of its medically diagnosed deficiency. It should be noted that there is a different regulation of medicines, which are subjected to marketing authorization procedure to assure their quality, safety, and efficacy, and food supplements, which are concentrated sources of nutrients or other substances with a nutritional or physiological effect and are regulated as foods. Therefore, the FDA-approved indications of vitamin B_12_ medicines include the following conditions: pernicious anemia, vitamin B_12_ malabsorption or dietary deficiency, atrophic gastritis, chronic use of acid-reducing medication, total or partial gastrectomy, small bowel bacteria overgrowth, infection with diphyllobothrium latum or helicobacter pylori, and pancreatic insufficiency [59]. All these conditions are associated with vitamin B_12_ deficiency and usually respond well to various vitamin B_12_ treatment strategies [60]. Intramuscular, oral (including sublingual), or intranasal applications have all been reported as effective routes for the treatment of vitamin B_12_ deficiency [60,61,62,63]. Irrespective of the cause, vitamin B_12_ deficiency is most commonly treated by intramuscular administration [13,64,65,66]. For example, in Spain, vitamin B_12_ deficiency is conventionally treated with 1000 μg CNCbl intramuscular injections, administered daily during the first week, followed by weekly administration for one month, and then every month or two months for life [66]. The recommended vitamin B_12_ form in the intramuscular injections and their treatment schedules differ from country to country, but treatment typically involves similar loading of doses at the beginning of the treatment, followed by maintenance treatment [39]. For comparison, the standard treatment of vitamin B_12_ deficiency without neurological symptoms in the UK is the intramuscular application of 1000 μg OHCbl three times a week for the first two weeks, followed by its application every 3 months until improvement, or for life in the case of pernicious anemia. The patients with neurological symptoms are treated with 1000 μg OHCbl intramuscular injections every other day until improvement, followed by a maintenance treatment every 2 months [13,53].

The application of high vitamin B_12_ doses is considered safe, as no adverse effects have been identified in association with its high intakes in both healthy individuals and patients with compromised vitamin B_12_ absorption. Vitamin B_12_ has also not been found to be carcinogenic and genotoxic in vivo or in vitro nor teratogenic or with adverse effects on fertility and post-natal development. Therefore, the EFSA Panel has not derived a tolerable upper intake level for vitamin B_12_ [1].

### 5.1. Different Routes of Vitamin B_12_ Application

The intramuscular route of vitamin B_12_ administration has gained wide acceptance due to its quite constant bioavailability (10% of the injected dose) in comparison with the variable bioavailability of the oral route, which depends on various transporters [39,60]. However, after the discovery of passive, transporter-independent vitamin B_12_ absorption, the effectiveness of its oral applications has been increasingly re-evaluated. Thus, high vitamin B_12_ doses (1000–2000 μg of CNCbl) are usually administered to ensure sufficient absorption to meet daily needs, even in the absence of transporter-mediated absorption [66,67,68,69,70,71]. The main conclusion of these studies, as well as of a Cochrane review on the efficacy of the oral versus the intramuscular route of vitamin B_12_ administration for the treatment of its deficiency, is that both application routes are similarly safe and effective in normalizing vitamin B_12_ serum concentrations [66,67,68,69,70,71,72]. These studies also highlight the main benefits of oral over the intramuscular route of vitamin B_12_ application, such as reduced patient injection-related discomfort and increased patient convenience, reduced risk of bleeding in anticoagulated patients, and reduced number of hospital visits [66,67,68,69,70,71,72]. In addition to these advantages, the use of orally administered vitamin B_12_ also results in a significant reduction in healthcare costs [64,65,73,74]. Therefore, some countries, such as Sweden and Canada, have implemented the routine use of oral vitamin B_12_ for deficiency treatment [75], and others, such as the UK, consider the use of oral vitamin B_12_ acceptable for the treatment of asymptomatic deficiencies, as a maintenance treatment, or for the prevention of vitamin B_12_ deficiency [53]. In addition to intramuscular and oral vitamin B_12_ administration, new routes, such as sublingual and nasal administration, have recently become available. In addition to the obvious advantages over the oral and intramuscular routes (patient convenience and good adherence, safety, cost-effectiveness), sublingual and nasal treatments also provide a promising alternative for specific populations (infants, children, and the elderly) and patients with some specific conditions (swallowing disorders or malabsorption due to intestinal surgery, inflammatory intestinal diseases, or short bowel syndrome), for which oral and intramuscular routes are less appropriate [60,63,76,77]. Their efficacy has been extensively evaluated in recent years and has been demonstrated to be comparable to those of oral or intramuscular vitamin B_12_ administration. Namely, the administration of sublingual vitamin B_12_ was shown to be comparable or even superior to the intramuscular route in infants [77,78], children [76,77], adults [60,79], and some specific population groups (patients with type 2 diabetes treated with metformin [80], and vegans and vegetarians) [81]. Similarly, nasal vitamin B_12_ has a demonstrated comparable efficacy to intramuscular vitamin B_12_ in children [61], adults [82], and the elderly [20,63,83].

### 5.2. Different Vitamin B_12_ Forms

CNCbl, the synthetic vitamin B_12_ form, has been traditionally used for the manufacture of vitamin B_12_ supplements and medicines, primarily due to its better stability compared to other vitamin B_12_ forms [3,84,85]. However, it is not a naturally present form of vitamin B_12_. It was first isolated from the liver, as an anti-pernicious anemia factor, due to an artefact of the purification method [84]. It is found only in trace amounts in human tissues as a result of cyanide binding produced by smoking or other sources [85]. More recently, the naturally occurring vitamin B_12_ forms—HCbl, MeCbl, and AdoCbl—have also become available for supplementation and therapy. These forms are often marketed and promoted with certain claims, including the superiority of supplemental AdoCbl in increasing B_12_ intramitochondrial levels or the superiority of supplemental MeCbl in intracellular methylation reactions, which are not necessarily scientifically based [85]. Some food supplements on the market contain edible blue-green algae (cyanobacteria), which typically contain pseudovitamin B_12_. However, this vitamin B_12_ form is inactive in humans [6].

All four authorized vitamin B_12_ forms are reported as being effective in improving its level in the human body [62,70,79,86,87,88,89]. As outlined above, before reaching their target cells, vitamin B_12_ is absorbed and transported by several vitamin B_12_ transporters, including HC, IF, and TC-II (Figure 1). Their affinity for different vitamin B_12_ forms is mainly reported as comparable [28,90,91,92,93]. Intracellularly, the free vitamin B_12_ form is released in the cytosol for its further activation (Figure 1). However, mutations in the genes encoding the lysosomal membrane protein LMBD1 result in an impaired vitamin B_12_ lysosomal release, its accumulation in the lysosomes, and reduced production of MeCbl and AdoCbl, resulting in methylmalonic aciduria and homocystinuria [94,95,96]. In this particular case, OHCbl was found to be superior to CNCbl in resolving the consequent neurological symptoms, suggesting its ability to bypass this classic lysosomal release mechanism [96,97]. As the free vitamin B_12_ form in the cytosol is further activated to its biologically active MeCbl and AdoCbl forms, these two forms seem to have a theoretical advantage over CNCbl and OHCbl, which require proper activation. However, numerous studies on vitamin B_12_ metabolism have shown that all four vitamin B_12_ forms are converted to the cob(II)alamin intermediate in the cytosol (Figure 1). During this process, which involves the activity of cytosolic MMACHC chaperon, assisted by methionine synthase reductase, flavins, NADPH, or reduced glutathione, the ligands of the specific supplemented vitamin B_12_ form—cyano, hydroxy, adenosyl, and methyl—are removed [28,85,98,99,100,101,102]. The cob(II)alamin intermediate is then converted into MeCbl in the cytosol or into AdoCbl in the mitochondrion (Figure 1). Thus, the amount of active vitamin B_12_ forms in the cells depends on the type of cell and on specific conditions, as well as on genetic polymorphisms, and not only on the supplemented vitamin B_12_ form [85,98]. The most common genetic disorder of intracellular vitamin B_12_ metabolism is Cobalamin C (CblC) deficiency, caused by mutations in the MMACHC genes [103,104]. It is characterized by impaired MeCbl and AdoCbl synthesis and impaired MMCA and MetH activity and clinically manifests as hyperhomocysteinemia, methylmalonic acidemia, and hypomethioninemia. The main treatment of CblC deficiency is OHCbl, which improves the biochemical parameters, hematological abnormalities, neuropsychiatric symptoms, and growth parameters [105,106,107,108]. The exact mechanism of CblC patients’ responses to OHCbl is still under investigation. Possible explanations include the non-specific, presumably glutathione-dependent, OHCbl reduction to the cob(II)alamin intermediate, causing OHCbl to be the only so-far reported vitamin B_12_ form that can bypass the MMACHC pathway [28]; and the higher binding affinity of OHCbl with respect to CNCbl for some MMACHC mutants related to CblC disorder [109].

Therefore, for the greater part of the population, the bioavailability of all four vitamin B_12_ forms—CNCbl, OHCbl, MeCbl, and AdoCbl—is considered comparable. However, the use of a specific vitamin B_12_ form, namely OHCbl, is advantageous in individuals with a genetic disorder of the intracellular vitamin B_12_ metabolism [28,85]. Moreover, there is a growing body of research demonstrating lower tissue retention, higher urinary excretion, and, consequently, lower overall bioavailability of vitamin B_12_, supplemented in the form of CNCbl, compared to OHCbl, MeCbl, and AdoCbl [85,110,111]. Additionally, some researchers suggest the use of natural vitamin B_12_ forms (OHCbl, MeCbl, or AdoCbl) instead of CNCbl for its long-term supplementations to avoid the accumulation of cyanide in human tissues, which is especially important for smokers, as tobacco smoke is one of the major sources of cyanide [85,112,113,114]. Consequently, there is an evident trend of the replacement of CNCbl supplements, which were previously almost exclusively found on the market, with its natural forms, especially MeCbl, presumably because of its cost-effectiveness [85].

## 6. Chemistry and Reactivity of Vitamin B_12_

Chemically, cobalamins consist of a corrin ring formed by four reduced pyrrole rings (A–D) and a central cobalt atom, chelated by the four pyrrole nitrogens and two additional ligands—nitrogen of the 5´,6´-dimethylbenzimidazole at the lower *α*-side and a ligand labeled R^1^ in Figure 3 at the upper *β*-side. Different vitamin B_12_ forms differ in the *β*-ligand and are listed in Figure 3. The fifth ligand-5´,6´-dimethylbenzimidazole- is linked to a ribose, which is connected to the corrin ring through a phosphate and propionamide group. Seven amide side chains (four propionamides and three acetamides) are also linked to the corrin ring [3,11,18,42,115,116,117]. Due to its complex structure comprising different functional groups, vitamin B_12_ is prone to various degradation reactions. The most typical degradation reactions in the corrin ring, its side chains, the nucleotide moiety, and the *β*-ligand are summarized below and are schematically presented in Figure 3.

Hydrolytic deamidation of the side chains

The amide (propionamide, shown in green, and acetamide, shown in blue, in Figure 3) side chains are susceptible to acid and alkaline hydrolysis of their associated carboxylic acids. The extent of the hydrolysis depends on the strength of the acid/base, time of exposure, and other exposure conditions (temperature, concentrations) and may be partial or complete (from mono- to heptacarboxylic acid). These hydrolytic degradation products are not biologically active [3,19,118,119].

Lactam formation

The acetamide side chain of the B ring in the structure of vitamin B_12_ is also susceptible to cyclization and the formation of a lactam in oxidative or alkaline conditions and elevated temperature. This reaction is indicated in purple in Figure 3. The formed compound, known as dehydrovitamin B_12_, has physical properties similar to CNCbl but without its biological activity [11,19,118,119].

Phosphate bond hydrolysis in the nucleotide moiety

Specific conditions (high acidity or presence of chromium(III) hydroxide and elevated temperature) cause the hydrolysis of the phosphate bond in the nucleotide moiety, yielding cobinamide and the benzimidazole nucleotide (labeled in pink in Figure 3), which is susceptible to further degradation [11,19,118].

Reduction in the cobalt atom

Chemical, electrochemical, or photochemical reduction in the central chelated cobalt atom (Co(III)) in the corrin ring of vitamin B_12_ may result in the formation of its Co(II) or Co(I) analogues, indicated as vitamin B_12_r and B_12_s, respectively (red color in Figure 3). Both of them are susceptible to oxidation and conversion into the Co(III) form [19,116,119,120]. As presented in Figure 3, the vitamin B_12_r form is easily oxidized into OHCbl in the presence of oxygen [11,19,121]. The formed OHCbl is in turn susceptible to further irreversible oxidative degradation [19,122].

Cleavage the of *β*-ligand Co-C bond

The cleavage and subsequent reformation of the Co-C bond in the vitamin B_12_ *β*-ligand occur both in vivo and in vitro as a consequence of external factors (light or elevated temperature) or the activity of Cbl enzymes in the body. These reactions are the basis for the biological activity of vitamin B_12_ [19,116,123]. The Co-C bond cleavage may be either homolytic or heterolytic. The homolytic cleavage, presented in red in Figure 3, is typical for AdoCbl and results in the formation of vitamin B_12_r and an adenosyl radical. The heterolytic cleavage is mostly catalyzed by MeCbl-linked enzymes and is, therefore, characteristic of MeCbl. It yields vitamin B_12_s with a lone electron pair, also known as “supernucleophile”, which further reacts with electrophiles (e.g., methylating agents) [19,116,117,120,123,124,125].

## 7. Vitamin B_12_ Stability

### 7.1. Factors Affecting the Stability of Different Vitamin B_12_ Forms

The intrinsic stability of vitamin B_12_ and the effects of various factors on its stability have wide reaching practical implications in the storage and management of B_12_-containing samples and they have been discussed for more than seven decades in the scientific literature. Initially, the research was focused on CNCbl, especially in the presence of ascorbic acid [126,127,128,129,130]. CNCbl is recognized as the most stable form of vitamin B_12_ and, as such, is still its most commonly used form for supplementation purposes [3,85,118,131,132]. However, with progress in the food, supplements, and pharmaceutical sectors and advances in stabilization approaches, there is a growing use of the other approved vitamin B_12_ forms (OHCbl, AdoCbl, and MeCbl) [85,133,134], resulting also in an increased interest in their stability. In general, considering Cbls’ structure and reactivity (Figure 3), we can conclude that they are susceptible to photolytic, hydrolytic, oxidative, and thermal degradation. Although some of these degradation reactions result in the formation of a biologically active form of vitamin B_12_ (e.g., formation of OHCbl after reductive decyanation of the cobalt atom in CNCbl, Figure 3), such conversions between different vitamin B_12_ forms in finished products, which specify the content of a single vitamin B_12_ form, are not acceptable in terms of their regulation. Thus, in the following, we provide a review of the stability of different vitamin B_12_ forms under different conditions, which, depending on the degradation mechanisms, considerably affect their stability.

#### 7.1.1. Temperature

Commercial vitamin B_12_ supplements (CNCbl, OHCbl, AdoCbl, and MeCbl) are typically supplied in dry solid forms (powders), with different recommended storage temperatures: 2–8 °C for CNCbl and H_2_OCbl^+^ as acetate, chloride, or sulfate and −20 °C for AdoCbl and MeCbl [135,136,137,138,139,140]. CNCbl is reported to be stable in its solid form even at elevated temperatures (up to 100 °C) for a few hours [3,117]. However, temperature is an important factor that strongly affects CNCbl stability in aqueous solutions. Temova et al. reported CNCbl degradation in aqueous solution after 24 h and 48 h of storage at 60 °C (5.3% and 8.6% degradation, respectively), which did not occur at room temperature (<1% degradation after 48 h) [141]. Bajaj and Singhal also showed that temperature significantly accelerates the CNCbl degradation rate in aqueous solutions with pH values between 2 and 10. The degradation follows first-order kinetics, with the half-life at pH 2 decreasing from 63 days at 4 °C to 8 days at 37 °C, while at pH 6 it decreases from 231 days to 116 days at the same temperatures. The maximum stability observed was at pH 6 and 4 °C with a halving of the lifetime at 37 °C. The high correlations between the degradation rate constants and the three evaluated temperatures (4, 25, and 37 °C) showed that CNCbl degradation followed the Arrhenius equation at tested solution aqueous media. The considerably different activation energy at pH 2 (41.9 kJ/mol) compared to the activation energies at pH 4–10 (between 10.3 and 15.4 kJ/mol) indicate different CNCbl degradation mechanisms at pH 2. The authors conclude that temperature has a greater effect on the stability of CNCbl in highly acidic conditions [142]. MeCbl is thermally stable in neutral aqueous solutions in the dark, as it persists after heating up to 100 °C for 20 min. However, the effect of temperature on MeCbl stability becomes evident in the presence of light, as it accelerates the photochemical transformation of MeCbl into OHCbl [3,86]. OHCbl is reported to be stable in a reconstituted solution for injection for up to 6 h at 40 °C [143]. However, the literature data regarding the comparative thermal stability of OHCbl are not fully consistent. Thus, Mander et al. report that the effect of temperature on the stability of OHCbl is comparable to that on CNCbl and MeCbl, leading to a similar extent of degradation under the same conditions [144], whereas Hadinata Lie et al. found that CNCbl is more heat stable than OHCbl and MeCbl [145]. Nonetheless, it can be concluded that the stability of CNCbl, MeCbl, and OHCbl in aqueous solutions is significantly affected by the temperature. The mechanism of thermolysis of AdoCbl is very well established. Homolysis dominated at high temperatures while heterolysis dominates at low temperatures [146]. The half-life at pH 7.5 changes from 690 days at 30 °C to 15 h at 85 °C [147]. The crystalline form is reported to decompose at temperatures >210 °C [3,86].

#### 7.1.2. PH of the Solution

Considering the liability of vitamin B_12_ to both acidic and alkaline hydrolysis, the effect of pH is important for its stability. Different vitamin B_12_ forms are generally unstable in both strongly acidic or alkaline media [3,117,148]. Among the different vitamin B_12_ forms, the most detailed information on the effect of pH on stability is available for CNCbl. Although some of the literature sources only highlight CNCbl’s instability in acidic conditions (pH < 3) [148], recent forced degradation studies showed an even greater extent of CNCbl degradation in strongly alkaline solutions. Specifically, CNCbl completely degraded in 0.1 M NaOH after 48 h of storage at room temperature (85% degradation after 1 h at 80 °C) in comparison with 11.2% degradation in 0.1 M HCl under the same conditions of CNCbl concentration, storage time, and temperature (61% degradation after 1 h at 80 °C) [141,149]. As reported above, Bajaj and Singhal evaluated CNCbl degradation kinetics at less extreme pH values (2, 4, 6, 8, and 10) at three temperatures (4, 25, and 37 °C) with the highest degradation extent and rate at pH 2 and lowest at pH 6 [142]. Other papers report the highest CNCbl stability in different pH ranges: 4–7 [3,117], 4–6.5 [150], 4.5–5 [151], and 4–4.5 [131]. The official requirements for the pH of CNCbl solutions are 4.0–5.0 for CNCbl oral solution in the British Pharmacopeia [152], and 4.5–7.0 for CNCbl injections in the United States Pharmacopeia [153]. To the best of our knowledge, pH profiles of the remaining vitamin B_12_ forms have not yet been published. However, the comparative stability of MeCbl and OHCbl with respect to CNCbl have been evaluated at pH 3.0, 4.5, 8.0, and 9.0 within 24 h of storage at room temperature. MeCbl degradation was most pronounced (78% at pH 3.0 and 64% at pH 9.0), followed by OHCbl (20% at pH 3.0 and 24% at pH 9.0) and CNCbl (16% at pH 3.0 and 15% at pH 9.0) [145]. Similar to CNCbl in the pH range 4–7, OHCbl, MeCbl, and AdoCbl are also reported to have the highest stability [145]. However, it should be considered that OHCbl is in equilibrium with its conjugate acid H_2_OCbl^+^. The corresponding logK of the following equilibrium equation H_2_OCbl^+^ + OH^−^ ⇄ OHCbl + H_2_O is 6.2 [154]. The official monographs specify a pH range for OHCbl preparations: between 3.8 and 5.5 for OHCbl injections in the British Pharmacopeia [152] and between 3.5 and 5.0 in the United States Pharmacopeia [153]. In this pH range, the H_2_OCbl^+^ is the prevalent form, however at neutral pH OHCbl is the main species.

#### 7.1.3. Exposure to Light

All four vitamin B_12_ forms, especially MeCbl and AdoCbl, are susceptible to photodegradation. The research in the field of vitamin B_12_ photostability goes back to 1956, with DeMerre and Wilson’s publication on the photolysis of CNCbl after exposure to different types of radiation. They reported that each hour of exposure to sunlight or UV light causes approximately 20% CNCbl degradation [155]. Since then, this field has been updated by numerous publications and is still being researched [156,157,158,159,160,161,162]. Vitamin B_12_ instability upon exposure to light is also recognized by the responsible authorities and manufacturers of vitamin B_12_ preparations, which recommend that vitamin B_12_ active ingredients (CNCbl, OHCbl, and MeCbl) and their preparations (injections and tablets) should be stored protected from light [140,152,153].

Different vitamin B_12_ forms undergo different photodegradation mechanisms. Upon light exposure, the cyanide ligand in CNCbl is replaced with an OH^−^ group to form OHCbl (in neutral or alkaline solutions) or with water to form H_2_OCbl^+^ (in acidic solutions) [122,163,164]. MeCbl and AdoCbl undergo photoreduction to vitamin B_12_r, which is quickly oxidized to OHCbl or H_2_OCbl^+^, depending on the pH [159,165,166]. Under the influence of light, OHCbl is irreversibly degraded to unknown corrin ring oxidation products [122,163].

In the literature, there are inconstancies regarding the comparative photostability of different vitamin B_12_ forms. Hadinata Lie et al. reported that CNCbl is significantly less susceptible to photolysis compared to OHCbl and MeCbl (4% CNCbl degradation after 60 min UV exposure compared to 28% for OHCbl and 39% for MeCbl in their aqueous solutions under the same experimental conditions) [145]. Juzeniene and Nizauskaite studied the photodegradation of different vitamin B_12_ forms in aqueous solutions at physiological pH (7.4) under UVA exposure. Under these conditions, OHCbl was the most stable vitamin B_12_ form, followed by CNCbl. The latter had around a five-fold higher photodegradation rate. MeCbl and AdoCbl were both particularly unstable and photodegraded within seconds of the light exposure. MeCbl was found to be three-fold more sensitive to UVA exposure than AdoCbl [164]. One of the possible reasons for the inconsistent results, apart from the different light sources, is the difference in the pH of the evaluated vitamin B_12_ aqueous solutions, which has been shown to significantly affect the rate of photodegradation. For example, Ahmad and Fareedi evaluated zero-order CNCbl photolysis in an aqueous solution across the pH range 1–12 and determined that the protonated CNCbl, present at lower pH, is more liable for photolysis (k = 1.316 × 10^−7^ M min^−1^ at pH = 1) than the neutral CNCbl form with the maximum photostability at pH = 8.5 (k = 0.039 × 10^−7^ M min^−1^) [161]. On the contrary, Vaid et al. showed that the protonated MeCbl was less susceptible to photolysis and that the first-order reaction rate increases with the pH up to pH 5.0 (k_obs_ = 0.134 min^−1^ at pH 1 and k_obs_ = 0.273 min^−1^ at pH 5) [165].

#### 7.1.4. Presence of Oxidizing and Reducing Agents

Vitamin B_12_ is unstable in the presence of both oxidizing and reducing agents [118,131,167,168]. Therefore, the instability of vitamin B_12_ in the presence of ascorbic acid, which was first discussed by Gakenheimer and Feller in 1949 [129], is the most known vitamin B_12_ incompatibility. Some reducing agents with the thiol functional group, including cysteine, homocysteine, N–acetyl–cysteine, cysteamine, pentafluorophenylthiolate, glutathione, and dithiothreitol form molecular complexes with vitamin B_12_ [163,169,170,171,172,173,174]. However, more in general, reducing agents, such as ascorbic acid, reducing sugars, thiols, formaldehyde, mercaptide ion, NaHSO_3_ or FeSO_4_, typically cause a reduction in the cobalt ion from Co(III) to Co(II), yielding vitamin B_12_r by the cleavage of the *β*-ligand [122,163,175,176,177]. In the presence of air, the vitamin B_12_r thus formed is oxidized to OHCbl [122,130]. Similar to the oxidation of other vitamin B_12_ forms, in some conditions (e.g., depleted oxygen concentrations and presence of hydrogen peroxide), vitamin B_12_r oxidation may also lead to irreversible cleavage of the corrin ring [118,122,178]. To ensure the oxidative stability of vitamin B_12_ active ingredients (CNCbl, OHCbl, and MeCbl) and preparations, the authorities recommend their storage in airtight containers [140,152,153].

**Ascorbic acid** is the most typical representative of reducing agents that react with vitamin B_12_ [122,126,127,129,130,179,180]. Its presence in solution is reported to cause up to 65% CNCbl degradation after 24 h of storage at 25 °C in the dark [122]. The destabilizing effect of ascorbic acid increases with the temperature, ascorbic acid concentration, and exposure of the solutions to light [122,179,181]. It also depends on the pH of the solution (faster reaction up to pH around 5.0 for both CNCbl and OHCbl) and on the vitamin B_12_ form (MeCbl is the least stable vitamin B_12_ form, followed by OHCbl and finally CNCbl) [122,145,179].

On the other hand, H_2_O_2_ is widely used as an oxidant, especially in forced degradation studies. Two published CNCbl forced degradation studies confirmed its oxidative instability following exposure to H_2_O_2_ [141,149]. However, the extent of CNCbl degradation in 3% H_2_O_2_ solution significantly differs between the two studies (~2% degradation after 24 h versus 55% degradation after 1 h at room temperature). Higher temperatures led to a significant increase in the rate and extent of CNCbl degradation in the presence of H_2_O_2_ [149].

### 7.2. Characteristic Interactions in Vitamin B_12_ Products

In addition to the above-described well-known interaction with **ascorbic acid**, the stability of vitamin B_12_ is also affected by the presence of some other compounds which are common ingredients of vitamin B_12_ preparations (foods or pharmaceutical preparations), such as other B-complex vitamins and some frequently used excipients. The most characteristic interactions are summarized below.

#### 7.2.1. Vitamin B_12_—Reducing Sugars (Dextrose and Sucrose)

Dextrose and sucrose are commonly used as excipients in both solid and liquid dosage forms, including parenteral preparations [182,183]. However, similar to the case of ascorbic acid, they have a reported incompatibility with CNCbl due to their reducing properties [183,184]. Barr et al. reported that, compared to sucrose, dextrose had a more pronounced effect on CNCbl degradation. There was a loss of about 40% in vitamin B_12_ content in the dextrose solutions and of nearly 30% in the sucrose solutions after 112 days at 45 °C. Dextrose was the only substance that produced an observable loss in vitamin B_12_ content at 25 °C. The deterioration at this temperature amounted to about a 25% loss in vitamin B_12_ content after 140 days [184].

#### 7.2.2. Vitamin B_12_—Riboflavin

Riboflavin is an effective photosensitizer that catalyzes the photochemical changes of other compounds, including also vitamin B_12_ [185,186,187]. Ahmad et al. evaluated the effect of riboflavin on CNCbl photostability in aqueous solutions in the pH range of 2–12. They concluded that riboflavin promoted CNCbl photolysis to OHCbl and further products, with a more than two-fold higher first-order reaction rate constant at pH 7 compared to the same solution in its absence. CNCbl photolysis in the presence of riboflavin showed the highest stability around pH 7–8 [188]. Similarly, Juzeniene and Nizauskaite evaluated OHCbl photodegradation in the presence of riboflavin at physiological pH (7.4) and concluded it was up to three-fold faster than its photodegradation in the absence of riboflavin [164].

#### 7.2.3. Vitamin B_12_—Thiamine

Vitamin B_12_ stability in aqueous solutions is also affected by the presence of thiamine or, more precisely, its degradation products. Feller and Macek as well as Mukherjee and Sen reported that CNCbl is stable in thiamine aqueous solutions at pH between 3.5 and 4.5 during prolonged storage at ambient temperature, but unstable under conditions that cause thiamine degradation (elevated temperature, high pH). They found the extent of CNCbl degradation related to thiamine degradation [189,190]. Doerge et al. identified the thiol-containing thiamine degradation product, cysteine, as responsible for CNCbl degradation [191]. This destabilization effect is enhanced by the presence of high thiamine concentrations and the presence of pyridoxine or nicotinamide [150,177,192].

#### 7.2.4. Vitamin B_12_—Nicotinamide

The adverse effect of nicotinamide on the stability of CNCbl in aqueous solutions has also been recognized by researchers for quite some time, with reports on their incompatibility since 1958–9 [190,193]. Ahmad et al. have more recently reported that nicotinamide accelerates CNCbl photolysis in an aqueous solution. They also evaluated the effect of pH in the range 1–7 and concluded that nicotinamide accelerates CNCbl photolysis across this pH range. In fact, pH significantly affects the rate of CNCbl photolysis in the presence of nicotinamide, which occurred as a second-order reaction (first-order in both species) at the highest rate at pH 1 (k_2_ = 0.57 M^−1^ min^−1^) and gradually decreased up to pH 7 (k_2_ = 0.075 M^−1^ min^−1^) [194].

#### 7.2.5. Vitamin B_12_—Mixtures of Water-Soluble Vitamins

Individual water-soluble vitamin deficiency is often associated with insufficient intake of other water-soluble vitamins. Therefore, they are commonly found all together in solid or liquid pharmaceutical preparation [141,195]. As described above, the presence of individual water-soluble vitamins (ascorbic acid, riboflavin, thiamine, and nicotinamide) affects vitamin B_12_ stability. Hadinata Lie et al. evaluated the stability of CNCbl, MeCbl, and OHCbl in the presence of both thiamine and nicotinic acid and showed that their presence causes the degradation of all three vitamin B_12_ forms, among which they identified MeCbl as the least stable form [145]. Temova et al. recently comparatively evaluated the stability of CNCbl in aqueous solutions alone and in multivitamin preparations (ascorbic acid, thiamine, riboflavin, riboflavin sodium 5′phosphate, nicotinamide, calcium-pantothenate, dexpanthenol, pyridoxine, biotin, and folic acid). They concluded that the presence of other water-soluble vitamins significantly decreased CNCbl stability under oxidative, thermal, and photolytic conditions, but did not affect its hydrolytic stability (in 0.1 M HCl and 0.1 M NaOH) [141].

## 8. Vitamin B_12_ in Foods, Food Supplements, and Medicines

### 8.1. Vitamin B_12_ Stability in Foods

Vitamin B_12_ is a natural component of certain foods (e.g., meat, milk, eggs, and fish) [6,7]. It is also commonly added to specific fortified foods (e.g., breakfast cereals, plant milk, and bread) in the food industry or formulated as medicines or food supplements in the pharmaceutical and food supplement industries [196,197]. In association with the general vitamin B_12_ instability, which, as described above, is affected by several factors, there are also various reports on its stability in foods and pharmaceutical preparations.

The effect of some typical food production processes (e.g., pasteurization, sterilization, and extrusion) on vitamin B_12_ content and stability has been examined primarily to evaluate possible food fortification with vitamin B_12_. Thus, Ottaway reported that vitamin B_12_ is stable during milk pasteurization. However, its stability in milk depends on the severity of the heat processing of the milk, with up to 20% and 35% loss during milk sterilization and spray drying, respectively [198]. Vitamin B_12_ (CNCbl and OHCbl) stability has also been evaluated throughout the processes of baking bread. Thus, Edelmann et al. concluded that the proofing step did not significantly affect CNCbl and OHCbl stability. Straight- and sponge-dough baking did not affect CNCbl content but caused an average of 21% and 31% OHCbl loss, respectively. The sourdough baking process caused both OHCbl (44%) and CNCbl degradation (23%) [199]. Extrusion has become a popular food processing method used for the production of foods, also vitamin-fortified, in various shapes and colors. Riaz et al., Killeit and Bajaj, and Singhal have evaluated how different extrusion variables affect CNCbl stability. Based on their research, we may conclude that CNCbl is susceptible to degradation during extrusion. Thus, the extent of CNCbl mostly depended on the die temperature (from 23% CNCbl loss at 140 °C to almost complete CNCbl degradation at 194 °C), whereas the screw speed and feed rate had a lesser effect on CNCbl stability [200,201,202]. In addition to the processes that vitamin B_12_ is exposed to during food production, the packaging of the foods is also reported to affect its stability. Hemery et al. determined CNCbl stability in fortified wheat flour packed in multilayer PET/aluminum bags or paper bags. The multilayer PET/aluminum bags, which are non-permeable to humidity and oxygen, provided sufficient CNCbl stabilization as no significant CNCbl loss was observed after 6 months of storage at 25 °C and 65% or 85% relative humidity (RH) or 40 °C and 65% or 85% RH. In contrast, the storage of the flour in paper bags at 65% RH resulted in a 30–48% and 49–63% loss of CNCbl after 1.5 and 6 months, respectively. Thus, packaging choice is of critical importance in CNCbl degradation in the flour samples [203].

Food processing has also been reported to significantly affect vitamin B_12_ stability, with a typical maximum vitamin B_12_ loss of up to 45% after cooking, 50% after cooking and draining, and 45% after reheating, as reported by Devi [204]. The effect of microwaving on vitamin B_12_ stability in foods was initially evaluated by Watanabe et al., who reported up to 40% total vitamin B_12_ loss after 6 min of microwave heating of beef, pork, and cow milk [205]. Similar conclusions were reached by Czerwonka et al. [206] and Bennink and Ono [207]. Nishioka et al. evaluated in greater detail how different cooking methods affect the stability of naturally present vitamin B_12_ in fish (round herring) meats. They reported a total vitamin B_12_ degradation of 59% after 7.5 min of grilling and 1.0 min of microwaving, 47% after 5.0 min of boiling, 43% after 4.0 min of frying, 41% after 9.0 min of steaming, and no vitamin B_12_ loss after 30 min of vacuum-packed pouch cooking. Thus, they concluded that vitamin B_12_ stability in food depends on the cooking temperature and time as well as the presence of other food ingredients, which was supported by a thermal stability study of OHCbl solutions [208]. Some specific food ingredients affecting vitamin B_12_ stability were identified by Johns et al. Specifically, they discovered that cocoa polyphenols (gallic acid, caffeic acid, epigallocatechin gallate, catechin, and its oligomers) accelerate CNCbl degradation, causing a three-fold greater vitamin B_12_ loss in heated, chocolate-flavored milk compared to unflavored milk [209]. The effect of the food matrix on vitamin B_12_ stability was also demonstrated by Bajaj and Singhal, who determined significantly higher CNCbl degradation rates in two model juices (lime and carrot) at 25 °C and 37 °C than in aqueous solutions with similar pH values. For instance, CNCbl degradation rate in lime juice (pH 2.16) was almost four-fold higher than in aqueous solutions with pH 2, which they presumed was a consequence of the high ascorbic acid concentration in lime juice. The role of the different complex matrices on CNCbl stability was evidenced by the different rate and extent of CNCbl degradation between the two juices evaluated (15% CNCbl loss after 28 days of storage at 25 °C in carrot juice compared to 95% loss in lime juice under the same experimental conditions) [142].

### 8.2. Vitamin B_12_ Stability in Food Supplements and Medicines

#### 8.2.1. Vitamin B_12_ Stability in Liquid Dosage Forms

Research on vitamin B_12_ stability in pharmaceutical preparations goes back to the 1950s, with Rosenblum and Woodbury’s publication on its stability in four different multivitamin combinations [210] and the report from Blitz et al. on its stability in commercial B-complex parenteral solutions [211]. The published results reveal incorrect vitamin B_12_ contents (between 5 and 80% of the labeled claim) in four of the six evaluated B-complex injectable solutions. After 3 months of storage at room temperature, an additional significant vitamin B_12_ loss was observed in all the tested preparations, resulting in its total degradation in half of the evaluated products and contents between 12 and 58% of the labeled claim in the remaining three solutions. Further storage at room temperature caused additional vitamin B_12_ degradation in all the evaluated preparations, except one. The authors observed a correlation between enhanced vitamin B_12_ degradation and higher thiamine and niacinamide concentrations [211]. Similar conclusions were drawn in a more recent study, performed by Ahmad and Hussain, evaluating CNCbl stability in seven commercial parenteral preparations. The initial CNCbl contents were in accordance with the labeled claims (101–110%) in three evaluated preparations that contained CNCbl only, in contrast to its low determined content (71–87%) in four tested injections that also contained thiamine and pyridoxine. A considerable difference in CNCbl stability between the single- and multi-ingredient preparations was generally observed. Thus, CNCbl contents remained at >95% of the labeled claim after 12 months of storage at room temperature in all three single-ingredient preparations, whereas its contents decreased to 28–37% of the labeled claim in the four multi-ingredient preparations [151]. CNCbl degradation in commercial parenteral preparations containing thiamine and pyridoxine, after two months of their light-protected storage at 40 °C, was additionally confirmed by Monajjemzadeh et al. By evaluating the CNCbl stability after 5 days at 55 °C in experimentally prepared parenteral formulations containing different CNCbl, thiamine, and pyridoxine combinations, they confirmed the CNCbl stability in a single-ingredient CNCbl solution as well as in combination with pyridoxine (97%) and its degradation in solutions with thiamine (79%) and especially in combination with thiamine and pyridoxine (41%) [150]. Recently, there is an increasing number of studies that highlight the photolability of MeCbl in liquid dosage forms and the need for its light-protected storage and proper stabilization [212,213,214].

#### 8.2.2. Vitamin B_12_ Stability in Solid Dosage Forms

In comparison with liquid dosage forms, the research on vitamin B_12_ stability in food supplements and medicines in solid dosage forms is less common. Jacob et al. first pointed out an unexpected vitamin B_12_ loss in film-coated multivitamin tablets and identified the use of methanol as a solvent for the film coating as a plausible cause for vitamin B_12_ degradation [215]. A more recent study, performed by Ohmoria et al., evaluated CNCbl stability in powders, granules, plain tablets, and sugar-coated tablets. They focused on the effect of humidity and the presence of other B-complex vitamins: fursultiamine (the disulfide derivative of thiamine), riboflavin, and pyridoxine. This study showed that moisture affects the CNCbl stability in powders, as the CNCbl powder remained stable after 4 months of storage at 40 °C/6% RH and 40 °C/57% RH but was unstable at 40 °C/75% RH. Compared to powders, CNCbl was more susceptible to moisture in granules that also contained fursultiamine, riboflavin, and pyridoxine. They also showed a negative correlation between water activity, expressed as equilibrium relative humidity (ERH), and CNCbl stability in plain tablets and sugar-coated tablets. Moreover, by comparison with uncoated tablets with the same ingredients and ERH levels, they demonstrated that the sugar-coating layer decreases the CNCbl stability. Evaluating the CNCbl stability in sugar-coated tablets with different compositions, they also showed that the presence of both riboflavin and pyridoxine reduced its stability, whereas fursultiamine, the tetrahydrofurfuryl thiamine derivative, improved its stability [216]. In addition to humidity and the presence of other interacting ingredients, light is also an important factor affecting the vitamin B_12_ stability in solid dosage forms. Its effects, as well as the importance of MeCbl light protection in solid dosage forms, were demonstrated by Saeki et al. Their six evaluated commercial MeCbl preparations in solid dosage forms (tablets and capsules) showed a different extent of MeCbl degradation (between 5 and 31% in the primary packaging and between 8 and 44% in the opened samples) after 20 days of light exposure, which depended on the packaging material and also the light-resistant properties of the products coating [217].

### 8.3. Vitamin B_12_ Content in Fortified Foods, Food Supplements, and Medicines and Their Regulation

The intrinsic instability of vitamin B_12_ in finished products is an important issue in ensuring their proper content-related quality. More specifically, its insufficient stabilization within formulations leads to its degradation during the production and storage of the finished products, which results in decreased content. Aware of this issue, manufacturers of products with such issues frequently adopt the strategy of the addition of sensitive active ingredients in higher contents than declared to compensate for degradation and therefore to guarantee the content and achieve a commercially acceptable shelf life [141,218,219,220]. Thus, this concept of overages is consistent with current Good Manufacturing Practices and is well recognized among the manufacturers of vitamin products [218,221,222]. The amount of used overage depends on the intrinsic stability of the active ingredient and its anticipated loss and should be reduced to a minimum [222]. To ensure food safety and the protection of consumers’ interests, the European Commission has published a guidance document for competent authorities for the control of compliance with EU legislation on the setting of tolerances for nutrient values declared on a label. The tolerances for vitamins, including vitamin B_12_ content, are between 65 and 150% in foods and between 80% and 150% in food supplements including measurement uncertainty. In the case of vitamin C in liquids, higher upper tolerances may be accepted [223]. In the USA, the U.S. Food and Drug Administration (FDA) regulates the lowest but not the highest vitamin content in fortified foods and food supplements. Thus, the FDA requires that Class I nutrients, which are nutrients (vitamins, minerals, protein, dietary fiber, or potassium) added in fortified or fabricated foods and food supplements, must be present at 100% or more of the value declared on the label; whereas Class II nutrients (vitamins, minerals, protein, total carbohydrate, dietary fiber, other carbohydrate, polyunsaturated and monounsaturated fat, or potassium) that occur naturally in food products must be present at 80% or more of the value declared on the label [224]. Tolerances for vitamin B_12_ content in food supplements are also provided in the United States Pharmacopeia-National Formulary (USP–NF) monographs, and are between 90 and 150% of its declared content on the label in capsules and tablets, whereas 90–450% of the labeled content is considered acceptable for oral solutions [153]. Vitamin B_12_ contents in medicines and the explanation of the acceptable tolerance are a part of the registration documentation and are directly regulated by the responsible regulation agency.

With the increase in health awareness, the aging population, lifestyle changes, and the global increasing cost of healthcare, the availability, and consumption of food supplements and fortified foods have been steadily increasing over the past decade worldwide [221,225]. Unfortunately, their growing use has not been accompanied by an increase in their quality. Due to variations in both manufacturing methods and regulatory controls, the actual contents of the active ingredients are unknown to researchers and consumers, who depend on product quality to ensure a sufficient intake to achieve the desired effects on one hand but avoid excessive consumption on the other [221,226,227]. Therefore, in line with the growing concern about their quality, there is also an increasing number of studies evaluating the contents of the active ingredients in fortified foods, food supplements, and medicines, including vitamin B_12_. To date, the most extensive study in this field was performed by Andrews et al., as a part of the analysis of multivitamin/mineral products in the United States market for the Dietary Supplement Ingredient Database. They analyzed a nationally representative range of multivitamin/mineral products, in which vitamin B_12_ was most often labeled in contents above its recommended Dietary Allowance (RDA), published by the US Institute of Medicine [228]. The determined vitamin B_12_ contents in the 98 tested food supplements had a labeled vitamin B_12_ content above its RDA between 50.6 and 164.3% of its labeled content, with an overall 8.4% mean percentage difference from the labeled content. The determined vitamin B_12_ content in the one additional food supplement with its labeled content below its RDA was 65.1% of the labeled content. The complete range of the final laboratory data, including 337 vitamin B_12_ food supplements, confirmed the wide distribution of its analytically determined contents, with the vitamin B_12_ content between 95% and 125% of the labeled contents in most of the tested supplements (25th–75th percentiles) [221]. The results of the remaining published studies evaluating the analytically determined content of vitamin B_12_ in relation to the labeled contents in fortified foods, food supplements, and medicines from around the globe were reviewed and are summarized in Table 1.

The summarized results (Table 1) provide an overview of the overall quality of vitamin B_12_ products. The published results of the vitamin B_12_ contents in fortified foods and food supplements are mostly spread between 100 and 150% of its labeled content, thus confirming the assumption that vitamin B_12_ overages are often added to commercial products. Nevertheless, the vitamin B_12_ contents were within the acceptable tolerances in the majority of the evaluated fortified foods (69%) and food supplements (87%). However, the available published data discloses vitamin B_12_ contents above the highest acceptable tolerance (150%) in a significant share of evaluated fortified foods (31%) and food supplements (11%), which reveal the need for more strict quality control of such products. The results on vitamin B_12_ contents in the evaluated medicines are also very concerning, especially in view of the significantly lower determined vitamin B_12_ content (<10% of the labeled content in three medicines and <80% of the labeled content in five additional medicines), which may be critical in the treatment of its deficiency. However, as the majority of these results on vitamin B_12_ content in medicines are outdated and published between 1956 and 2000, this field should be updated with some more recent data to draw firmer conclusions.

## 9. Strategies for Vitamin B_12_ Stabilization

Proper vitamin B_12_ stabilization and content in relation to the values declared on a label is very important in ensuring the quality, efficacy, and safety of vitamin B_12_ preparations. Therefore, a part of the scientific community has recently focused on researching possibilities for vitamin B_12_ stabilization, most of which are based on the minimization of the destabilizing factors, the addition of proper stabilizers, or the application of some technological processes to enhance its stability. Considering the aforementioned hydrolytic, thermal, oxido-reductive, and photolytic instability, vitamin B_12_ can be significantly stabilized by adjusting the pH of the solution to 4–7, storing it at a lower temperature, protecting it from light, oxidants, and reductants, and formulating it into solid dosage forms [145,150,161,164].

The most recognized additives that stabilize vitamin B_12_ include sorbitol, ferric salts (ferric chloride and saccharated iron oxide), whey proteins (alpha-lactalbumin, beta-lactoglobulin, and lactoferrin), and halide salts (potassium, magnesium, and calcium halides). The protective effect of **sorbitol** on the stability of vitamin B_12_ was first mentioned by Barr et al. (1957), which was explained by its ability to decrease the quantity of available water that promotes its hydrolysis [184]. After a research intermission, the stabilizing effect of sorbitol on CNCbl, MeCbl, and OHCbl in solutions and medical food has been recently evaluated in more detail. Hadinata Lie et al. showed that sorbitol significantly reduced the extent of photodegradation of CNCbl (from 3.8% to 0% degradation loss after 60 min of exposure to UV light), OHCbl (from 28% to 10% loss), and especially MeCbl (from 39% to 1% loss) in solutions. Similarly, they showed that the addition of sorbitol to their solutions significantly improved the thermal stability of all three evaluated vitamin B_12_ forms (from 38% to 1% degradation loss of CNCbl, from 34% to 10% for OHCbl, and from 38% to 20% for MeCbl after 60 min at 100 °C). The stability of CNCbl, OHCbl, and MeCbl in the presence of ascorbic acid or both thiamine and nicotinic acid also significantly improved in the presence of sorbitol. Moreover, sorbitol effectively protected CNCbl and MeCbl against acidic and alkaline hydrolysis but did not have a sufficient stabilizing effect on the hydrolysis of OHCbl [145]. Sorbitol, being a polyol sweetener with an antihyperglycemic effect, is a common ingredient in the food and pharmaceutical industry, including medical food for diabetic patients. Therefore, Lee et al. evaluated its adequacy as a CNCbl stabilizing agent during the production process of medical food. In addition to its protective effect on the thermal and ascorbic acid-induced degradation of CNCbl in solutions, Lee et al. also demonstrated that sorbitol efficiently protected CNCbl against degradation in medical foods, caused by heat (two-step sterilization) and the food matrix, which also contained ascorbic acid [181].

Similar to the case of sorbitol, the stabilizing effect of **ferric chloride** on vitamin B_12_ stability in the presence of reducing agents, first mentioned by Mukherjee in 1959 [190], was recently explored in more detail by Lee et al. [181]. The protective effect of ferric chloride is based on its oxidizing ability, which protects the B_12_ from reducing agents such as ascorbic acid [142,181,190]. Lee et al. showed that ferric chloride reduced CNCbl degradation, caused by the combination of ascorbic acid and heat, to a minimum. However, they concluded that despite its protective effects, ferric chloride is less appropriate than sorbitol for CNCbl stabilization in commercial products because of its catalyzing effect on the oxidation of ascorbic acid, which is a common ingredient of multivitamin products. In addition, there are limitations on its added amounts due to the recommended daily intake of chlorine and iron [181]. Ichikawa et al. showed that the incompatibility of vitamin B_12_ and ascorbic acid can be also overcome by the addition of various halide salts. They demonstrated that the presence of potassium, magnesium, and calcium halides improves the stability of both CNCbl and ascorbic acid in the pH range of 3.5–5.3 and that their stabilization effect increases with an increase in the concentrations and the atomic number of the halide anion Cl^−^ < Br^−^ < I^−^) [242].

Vitamin B_12_ complexation with **whey proteins** (alpha-lactalbumin, beta-lactoglobulin, and lactoferrin) is also an effective approach for its stabilization. Wang et al. evaluated the effect of alpha-lactalbumin and beta-lactoglobulin on the thermal and photostability of CNCbl and AdoCbl and showed an increase in their stability by 10–30%. Thus, they showed that these whey proteins improve the stability of CNCbl and AdoCbl upon exposure to heat and light during food processing and also during in vitro stomach digestion, leading to their improved bioavailability [243]. Similarly, Uchida et al. patented the production of the CNCbl-lactoferrin complex, in which lactoferrin provided a considerable increase in CNCbl stability against acidic hydrolysis. Due to the high photostability of the thus prepared complex, the innovators claim its suitability for widespread use in foods and drinks fortification and the production of medicines [244].

Researchers have also responded to the high demand for effective vitamin B_12_ stabilization for fortification and supplementation purposes by exploring **technological approaches** for improving its stability and compatibility with other, commonly present ingredients, such as other vitamins and, particularly, ascorbic acid or minerals. Such strategies include the use of viscosigens, co-crystallization and microencapsulation techniques, vitamin B_12_ incorporation in bilayer tablets, liposomal preparations, and nanosystems [245,246,247].

Grissom et al. reported that CNCbl can be significantly stabilized against photolysis in anaerobic conditions (aqueous solutions purged with nitrogen or argon for more than 30 min and sealed with a rubber septum) by the use of **viscosigens** such as glycerol (≥25%) or Ficoll^®^, which is a hydrophilic copolymer of saccharose and epichlorohydrin (≥10%). They propose the decrease in diffusion and enhancement of radical-pair recombination as the likely mechanism for CNCbl stabilization by viscosigens [245].

Bajaj and Singhal approached the stabilization of CNCbl using **co-crystallization**, which is a previously reported and widely used technique for the stabilization of different sensitive compounds in the food industry, including flavor compounds, antioxidants, and some vitamins (K_3_ and D_3_) [246,248,249]. They reported that the CNCbl co-crystallization with sucrose and gum acacia, as a hydrocolloid (2.5 g/100 g sucrose), results in substantial CNCbl stabilization, which is additionally enhanced by storage at lower temperature and RH (25 °C/33% RH). At these storage conditions, the authors report a low first-order degradation rate constant (0.001 day^−1^) and a half-life of 693 days. The authors conclude that co-crystallization is a simple, easily adoptable, and cost-effective approach for CNCbl stabilization and its food fortification [246].

The development of a stable food supplement formulation, containing CNCbl in combination with all other B-complex vitamins (thiamine mononitrate, riboflavin, nicotinamide, calcium pantothenate, pyridoxine hydrochloride, folic acid, and biotin), as well as vitamins C (ascorbic acid), A (retinyl acetate), D (cholecalciferol), and E (tocopherol) and the following minerals: copper, iodine, magnesium, manganese, molybdenum, silicon, and zinc, was addressed by Rajakumari et al. [247]. In this recent research, they pointed out the disadvantages of conventional food supplement formulations (film-coated and sugar-coated tablets), including decreased stability of the active ingredients and their interactions with other ingredients and highlighted the need for the development of contemporary food supplement formulations with improved properties. They described the formulation of a **film-coated bilayer tablet** using a direct compression method in which vitamins are incorporated in the first and minerals in the second layers. By optimizing the composition of the bilayer tablet, through the inclusion of microcrystalline cellulose, carmellose calcium, sodium starch glycolate, croscarmellose sodium, and pregelatinised starch as excipients, they achieved appropriate physicochemical stability for all the incorporated vitamins and minerals. The authors claim the determined shelf-life (≥ 90% of the initial content of all the active ingredients) of 24 months increased production and time- and cost-effectiveness as the main benefits of the proposed approach for its applicability in the food and pharmaceutical industry [247].

Another of the explored approaches for vitamin B_12_ stabilization is based on the particle size reduction and includes microencapsulation, nanoencapsulation, and the production of liposomes. **Microencapsulation** is an advanced and emerging technology in the food and pharmaceutic industry for the protection of unstable food components or active ingredients by their incorporation into particles of micrometer size [250,251]. Several techniques, including spray-drying [252,253], spray-chilling [254], and emulsion technique [255] have been reported for CNCbl microencapsulation. All these authors report that by optimization of the microencapsulation process and composition, high CNCbl encapsulation efficiency and stability can be achieved, causing this technique to be a promising solution for the development of quality and stable vitamin B_12_ products. For instance, Carlan et al. incorporated CNCbl into modified chitosan microparticles and achieved a ≤ 10% CNCbl loss after 6 months of storage at 25 °C [253], while Mazzocato et al. report a ≤ 5% loss of CNCbl incorporated into solid lipid microparticles after 4 months of storage at 25 °C, compared to the ≈25% loss of its free form under the same conditions [254]. Further reduction in the particle size to the nano-scale and the use of **nanocarrier-based delivery systems** is another promising approach for the improvement of vitamin B_12_ stability [256]. In that sense, Britto et al. describe the encapsulation of CNCbl into polymeric nanoparticles, formed by chitosan and tripolyphosphate. Under the tested conditions, the encapsulated and non-encapsulated control (aqueous CNCbl solution (pH 6.8) at 25 °C, in the dark for three weeks) remained stable [257]. Unfortunately, the conditions used did not allow for the observation of the stabilizing effect of B_12_ nanoencapsulation, while the stabilization effect of nanoencapsulation was observed for vitamin C. Maiorova et al. emphasized the applicability of nanosystems as carriers for CNCbl by the development and evaluation of nanoengineered polymer capsules and lyotropic liquid-crystalline nanosystems [258]. The incorporation of vitamin B_12_ into **liposomes** has also been shown as an effective approach for its protection from the effect of destabilizing factors [259,260,261]. Liposomes are closed spherical vesicles with a unique structure, composed of a phospholipid bilayer, enabling vitamin B_12_ incorporation inside the center aqueous phase, which protects it from degradation. This approach was utilized by Arsalan et al., who investigated the effect of CNCbl entrapment in liposomal preparations on its photostability. They determined a two-six-fold lower photolysis rate of CNCbl in liposomes, compared to the unentrapped CNCbl under the same conditions. This result can be further improved by alteration in the used phospholipid (phosphatidylcholine) and the size of the liposomes [259]. Similarly, Lee et at. demonstrated that CNCbl incorporation into ordinary or chitosan-coated liposomes enhances its stabilization in the presence of ascorbic acid. The stabilization effect was more evident at higher storage temperatures and can be exploited for the production of quality food supplements containing both CNCbl and ascorbic acid [261].

## 10. Conclusions

Vitamin B_12_ has been recognized as an essential micronutrient for almost one century. In the form of medicines, it is mainly used for the treatment of vitamin B_12_ deficiency, whereas vitamin B_12_ supplements (including fortified foods) are commonly used for the prevention of its deficiency, especially in high-risk population groups (vegetarians, vegans, the elderly, pregnant, and lactating women). Four vitamin B_12_ forms—CNCbl, OHCbl, AdoCbl, and MeCbl are authorized for supplementation purposes. The overview of the chemical, biochemical, and pharmacological aspects of these vitamin B_12_ forms, reported in this review, provides the necessary background to understand the factors affecting the stability of this micronutrient in simple and complex matrices. CNCbl has been traditionally used in vitamin B_12_ supplements and medicines, primarily due to its superior stability compared to other vitamin B_12_ forms. Thus, it is also the most researched form both in terms of efficacy and stability, but its use is recently on the decline, due to concerns about the potential accumulation of cyanide in human tissues. In that sense, MeCbl, OHCbl, and AdoCbl could be more appropriate and promising vitamin B_12_ forms. However, their intrinsic lower stability is an obstacle that needs to be overcome for their wide use for vitamin B_12_ supplementation in clinical practice. Stability is one of the main aspects in ensuring the quality, efficacy, and safety of the vitamins. Based on the present review of the literature, we conclude that research on strategies for vitamin B_12_ stabilization is principally limited to CNCbl. However, considering the inferior stability of AdoCbl, OHCbl, and MeCbl compared to CNCbl, such research would be more valuable for these three vitamin B_12_ forms. In any case, we believe that this extensive review of the approaches for stabilization of different vitamin B_12_s will provide researchers, manufacturers of fortified foods, food supplements, and medicines, and interested professionals with an overview of this issue and contribute toward an increase in the quality of vitamin B_12_-containing products.

## Figures and Tables

**Figure 1 molecules-28-00240-f001:**
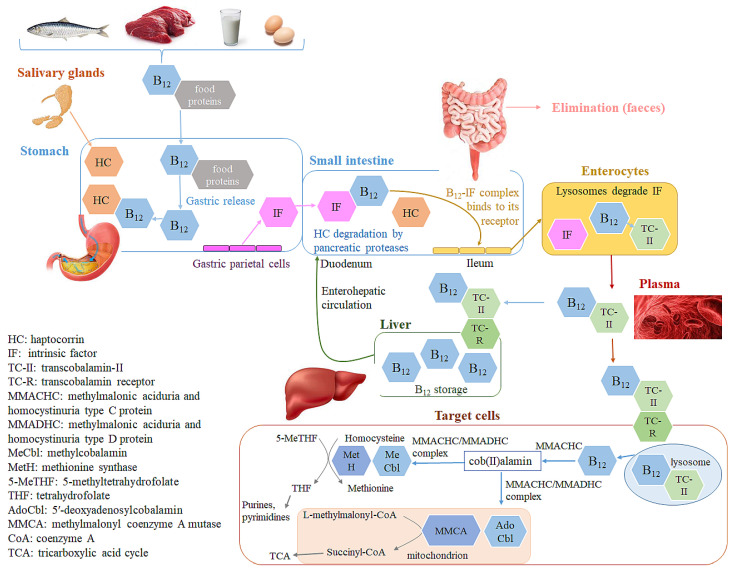
Schematic representation of vitamin B_12_ absorption, distribution, metabolism, excretion (ADME), enterohepatic circulation, and cellular uptake and activity.

**Figure 2 molecules-28-00240-f002:**
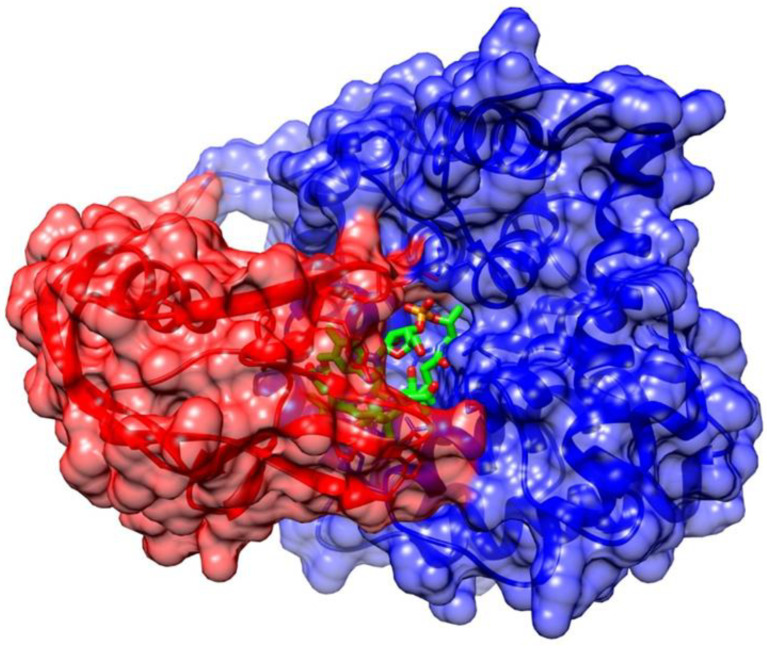
Representation of TC-II protein structure [35]. Secondary structure cartoon showing the N-terminal α-domain (blue), the C-terminal β-domain (red), and the Cbl (green) in the central domain interface. The protected Cbl is tightly encapsulated between the two protein domains.

**Figure 3 molecules-28-00240-f003:**
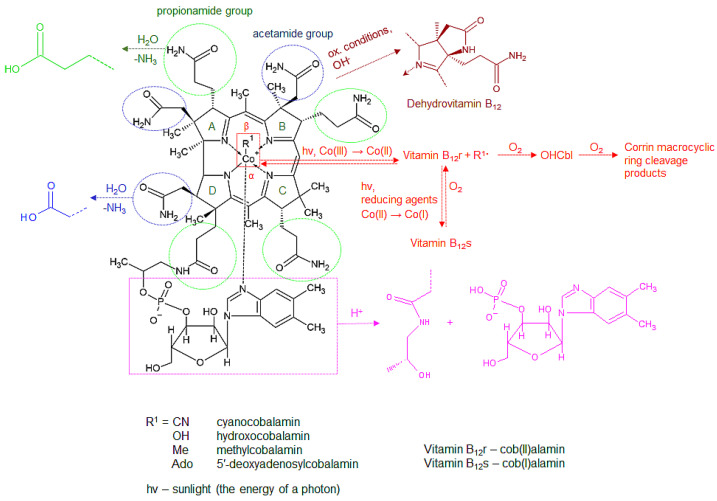
Typical degradation reactions of vitamin B_12_.

**Table 1 molecules-28-00240-t001:** The compliance of the determined vitamin B_12_ contents in the tested fortified foods, food supplements, and medicines with the label claims along with the lowest (Min), highest (Max), and average (AV) determined contents and the number of tested products in each category (No.). The pie chart represents the shares of products with determined vitamin B_12_ content below (blue), within (green), and above (red) the acceptable tolerances for each product category (65–150% in fortified foods, 80–150% in food supplements, and 90–110% for medicines).

Vitamin B_12_ Content	Fortified Foods	Food Supplements	Medicines
**AV (Min–Max) (%)**	131.4 (85.4–242.5)	115.6 (56.0–281.8)	85.9 (5.0–178.0)
	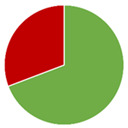	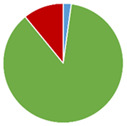	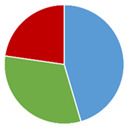
**No.**	13	46	22
**References**	[229,230,231,232]	[133,141,232,233,234,235,236,237,238,239,240]	[141,150,151,211,235,237,241]

## Data Availability

Not applicable.

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
