# Peer review of "Vitamin B12 in Foods, Food Supplements, and Medicines—A Review of Its Role and Properties with a Focus on Its Stability"

_molecules, 2022, doi:10.3390/molecules28010240_

Round 1

Reviewer 1 Report

In this review article the authors greatly described each and every aspect related to Vitamins. As we know that Vitamin B12 is a widely used compound in the feed and food, healthcare and medical industries that can only be produced by fermentation because of the complexity of its chemical synthesis. For this reason, finding better producer strains and optimizing their bioprocesses have been the main focus of industrial producers over the last few decades. This review article covered all the aspects.

This review article is the perfect addition to the field; it is very well written with detailed information and I haven’t seen any gap.

In Current state, CNCbl production is still suboptimal and has many challenges to overcome to further develop its potential as a cost-effective and valuable industrial bioprocess.

Regarding the methodology, further efforts in bioprocessing, downstream and media composition optimization (with cheaper or recycled compounds) should be carried out to increase the economic viability and environmental sustainability of vitamin B12 biotechnological production. However, the main problem, still, is the low productivity of the available producing strains, caused mainly by the tight genetic regulation of Cbl production: the inhibition of the cysG and the cbi operon by the cobalamin riboswitch, as well as other down regulating processes.

The conclusions consistent with the evidence and arguments presented but Overcoming this limitation may require genetic engineering, which may not be well received by end consumers, mainly vegans or vegetarians, who are very concerned about their diet choices and the usage of GMO organisms.

Very well written and representation. The figures are perfectly made. I would recommend the publication of this article in present form.

Author Response

We thank the referee for his/her positive appreciation of our work and useful comments.

Reviewer 2 Report

The manuscript "Vitamin B12 in Foods, Food Supplements, and Medicines—A Review of its Role and Properties with a Focus on its Stability" provides a nice overview of the pharmacological, chemical and biochemical properties of a complex vitamin. The review is well written and work is highly significant. A few suggestion for the authors: 

Figure 1) The figure depicts MMADHC as involved in the processing of B12 from MMACHC to MetH and MMCA. The process is more complex and suggest changing the figure to indicate the MMACHC/MMADHC complex.

Author Response

We thank the referee for his/her positive appreciation of our work and helpful comments which have improved our manuscript.

In particular, Figure 1 has been changed to take into account the referee’s suggestion.

Reviewer 3 Report

Abstract: Delete “also known as the anti-pernicious anemia factor”? PA is only a small part of its usefulness. And this reviewer never hears it called that.

L15. DependS.

Page 1 and throughout. How is a “B12 medicine” defined separately from a supplement?

Line 111. (after injection)

Figure 1. “ADME” needs adding to the list of abbreviations

Line 145. Serum and urine

Section 4. Another vulnerable group is young infants consuming milk from mothers with poor vitamin B12 status, resulting in low milk concentrations.

Line 180. B12 deficiency has to be quite severe before haematological changes occur, and this should be noted. See the NIH BOND report on vitamin B12 https://www.ncbi.nlm.nih.gov/pmc/articles/PMC6297555/

Line 204. The concept of B12 in pill form being called a “food supplement” is unusual. “Food supplement” usually means addition of foods high in B12 or fortified foods. The medical conditions then listed are also commonly treated with supplements. Again, the term “vitamin B12 medicine” is strange. A different terminology is likely needed – e.g. high dose, or injectable etc. Sublingual is also oral.

Line 276. First

Figure 3. Add “hv” to list of abbreviations

Section 7. Link the facts I this section with their real world practical implications e.g. for blood collection in the field, sample storage and management.

Line 950. Improved.

Line 959. Where does fortification fit in your dichotomous definition?

Ref 61 needs editing.

Author Response

We thank the referee for his/her positive evaluation and the helpful comments which have improved our manuscript.

The detailed responses to the specific comments and the changes introduced in the revised version of the manuscript are reported below:

REVIEWER: Abstract: Delete “also known as the anti-pernicious anemia factor”? PA is only a small part of its usefulness. And this reviewer never hears it called that.

REPLY: The unknown anti-pernicious anemia factor, independently isolated from liver in 1948 by two industrial laboratories, namely, those of Merck and Co. (New Jersey, USA) and Glaxo Laboratories (Middlesex, Great Britain) was successively call Vitamin B12 by the American group see: Anti-Pernicious Anæmia Factor. Nature 161, 676 (1948). (https://doi.org/10.1038/161676a0). The historical origin of the name is mentioned in the introduction of the paper.

REVIEWER: L15. DependS.

REPLY: Corrected

REVIEWER: Page 1 and throughout. How is a “B12 medicine” defined separately from a supplement?

REPLY: The authors recognize that the distinction between medicines and supplements could be, sometimes, ambiguous. Therefore, we have modified the text in order to clarify that we intend the vitamin B12 in the form of “prescribed medicines” which may not be dispensed legally without the prescriptions of a qualified medical practitioner. The distinction between medicines and supplements is important because there is a different regulation of medicines and food supplements (see below).

REVIEWER: Line 111. (after injection)

REPLY: Corrected

REVIEWER: Figure 1. “ADME” needs adding to the list of abbreviations

REPLY: The legend of Figure 1 has been updated with “absorption, distribution, metabolism, excretion (ADME)”

REVIEWER: Line 145. Serum and urine

REPLY: "and Urine" has been included in the text

REVIEWER: Section 4. Another vulnerable group is young infants consuming milk from mothers with poor vitamin B12 status, resulting in low milk concentrations.

REPLY: The category “infants breastfeeding from mothers with B12 deficency” has been included in the revised manuscript

REVIEWER: Line 180. B12 deficiency has to be quite severe before haematological changes occur, and this should be noted. See the NIH BOND report on vitamin B12 https://www.ncbi.nlm.nih.gov/pmc/articles/PMC6297555/

REPLY: The diagnosis of “severe” vitamin B12 deficiency has been corrected.

REVIEWER: Line 204. The concept of B12 in pill form being called a “food supplement” is unusual. “Food supplement” usually means addition of foods high in B12 or fortified foods. The medical conditions then listed are also commonly treated with supplements. Again, the term “vitamin B12 medicine” is strange. A different terminology is likely needed – e.g. high dose, or injectable etc. Sublingual is also oral.

REPLY: We have changed the following sentence: “Contrary to dietary supplements, which are generally used for the prevention of vitamin B12 deficiency, high dose vitamin B12 treatments are indicated and used in conditions of its medically diagnosed deficiency. It should be noted that there is a different regulation of medicines, which are subjected to marketing authorisation procedure to assure their quality, safety and efficacy, and food supplements, which are concentrated sources of nutrients or other substances with a nutritional or physiological effect, and are regulated as foods.” Furthermore, the text has been modified: “oral (including sublingual)”

REVIEWER: Line 276. First

REPLY: Corrected

REVIEWER: Figure 3. Add “hv” to list of abbreviations  

REPLY: Actually “hν” is not an abbreviation but it is the product between the Plank constant (h) and the electromagnetic frequency (ν) ad it represents the well-known photon energy. It is a well-established way to indicate a photochemical reaction. The figure 3 has been modified to include this information.

REVIEWER: Section 7. Link the facts I this section with their real world practical implications e.g. for blood collection in the field, sample storage and management.

REPLY: The initial sentence of section 7 has been updated: “The intrinsic stability of vitamin B12 and the effects of various factors on its stability have wide reaching practical implications in storage and management of B12 containing samples and they have been discussed for more than seven decades in the scientific literature.”

REVIEWER: Line 950. Improved.

REPLY: Corrected

REVIEWER: Line 959. Where does fortification fit in your dichotomous definition?

REPLY: The sentence has been modified to explicitly include the fortified foods: “…whereas vitamin B12 supplements (including fortified foods) are commonly used….”

REVIEWER: Ref 61 needs editing.

REPLY: Reference 61 has been corrected as well as other references